



# Captive Aerosol Growth and Evolution (CAGE) chamber system to investigate particle growth due to secondary aerosol formation

Candice L. Sirmollo[1,2], Don R. Collins[1,2], Jordan M. McCormick[3], Cassandra F. Milan[3], Matthew H. Erickson[4], James H. Flynn[4], Rebecca J. Sheesley[5], Sascha Usenko[5], Henry W. Wallace[6], Alexander A. T. Bui[6], Robert J. Griffin[6], Matthew Tezak[7], Sean M. Kinahan[7,8], Joshua L. Santarpia[9]

[1]Department of Chemical and Environmental Engineering, University of California Riverside, Riverside, California 92521 USA
[2]College of Engineering, Center for Environmental Research and Technology (CE-CERT), University of California Riverside, Riverside, California 92507 USA
[3]Department of Atmospheric Sciences, Texas A&M University, College Station, Texas 77843 USA
[4]Department of Earth and Atmospheric Sciences, University of Houston, Houston, Texas 77204 USA
[5]Department of Environmental Science, Baylor University, Waco, Texas 76798 USA
[6]Department of Civil and Environmental Engineering, Rice University, Houston, Texas 77005 USA
[7]Sandia National Laboratory, Albuquerque, New Mexico 87123 USA
[8]Biodefense and Health Security, University of Nebraska Medical Center, Omaha, Nebraska 68198 USA
[9]Department of Pathology and Microbiology, University of Nebraska Medical Center, Omaha, Nebraska 68198 USA

*Correspondence to*: Don R. Collins (donc@ucr.edu)

**Abstract.** Environmental chambers are a commonly used tool for studying the production and processing of aerosols in the atmosphere. Most are located indoors and most are filled with air having prescribed concentrations of a small number of reactive gas species. Here we describe portable chambers that are used outdoors and filled with mostly ambient air. Each all-Teflon® 1-m³ Captive Aerosol Growth and Evolution (CAGE) chamber has a cylindrical shape that rotates along its horizontal axis. A gas-permeable membrane allows exchange of gas-phase species between the chamber and surrounding ambient air with a mixing time constant of approximately 0.5 h. The membrane non-permeable to particles, and those that are injected into or nucleate in the chamber are exposed to the ambient-mirroring environment until being sampled or lost to the walls. The chamber and surrounding enclosure are made of materials that are highly transmitting across the solar ultraviolet and visible wavelength spectrum. Steps taken in the design and operation of the chambers to maximize particle lifetime resulted in averages of 6.0 h, 8.2 h, and 3.9 h for ~0.06 μm, ~0.3 μm, and ~2.5 μm diameter particles, respectively. Two of the newly developed CAGE chamber systems were characterized using data acquired during a 2-month field study in 2016 in a forested area north of Houston, TX, U.S. Estimations of measured and unmeasured gas-phase species and of secondary aerosol production in the chambers were made using a zero-dimensional model that treats chemical reactions in the chamber and the continuous exchange of gases with the surrounding air. Concentrations of NO, NO₂, NOᵧ, O₃, and several organic compounds measured in the chamber were found to be in close agreement with those calculated from the model, with all having near 1.0 best fit slopes and high r² values. The growth rates of particles in the chambers were quantified by tracking the narrow modes that resulted from injection of monodisperse particles and from occasional new particle formation bursts. Size distributions in the two chambers were measured intermittently 24 h day⁻¹. A bimodal diel particle growth rate pattern was observed, with maxima of about 6 nm h⁻¹ in the late morning and early evening and minima of less than 1 nm h⁻¹ shortly before sunrise and sunset. A pattern change was observed for hourly averaged growth rates between late summer and early fall.



## 1 Introduction

Atmospheric aerosols play a role in atmospheric chemistry, health effects, and climate forcing. Secondary aerosol is produced in
the atmosphere from the oxidation of precursor gas-phase species and can either add to existing particles or nucleate to form new
ones, which is the initial step in the process known as new particle formation (NPF). Condensation, gas-particle partitioning, and
heterogeneous reactions are known mechanisms by which secondary aerosol contributes to particle growth. Several groups have
developed particle growth models that are constrained by measurements and that represent some or all of these growth mechanisms
(Stolzenburg et al., 2005; Yli-Juuti et al., 2013; Tröstl et al., 2016). Atmospheric aerosol formation and growth is regionally and
globally significant and thus these modeled mechanisms should be included in larger-scale models that investigate the climate
effects of aerosols (Kulmala and Kerminen, 2008). Though aerosol formation, growth, and atmospheric processing have been the
focus of many studies, further investigations are required to realistically represent aerosol behavior (Kroll and Seinfeld, 2008;
Hallquist et al., 2009; Laj et al., 2009).

Laboratory reactors such as environmental chambers and oxidation flow reactors are tools that are commonly used to better
understand and predict atmospheric processes under controlled settings. Environmental chambers have been used in the laboratory
and the field to study gas-phase kinetics, urban air pollution, particle formation and growth, and aqueous secondary organic aerosol
(SOA) production. They have been used to investigate secondary aerosol formation from vehicle exhaust (Weitkamp et al., 2007;
Vu et al., 2019), physical, chemical, and optical properties of aging biomass burning particles (Hennigan et al., 2011; Zhong and
Jang, 2014; Tkacik et al., 2017; Smith et al., 2019) and the impact of atmospheric conditions on the viability of bacteria (Brotto et
al., 2015; Massabò et al., 2018). They provide a method to simulate the aerosol production that would occur in a parcel of air in
the atmosphere. Interpretations of gas-particle interactions in chamber systems can guide development of models and model
parameterizations used to describe real-world atmospheric processing of particles.

Environmental chambers vary in design in their volume, materials, light source, and temperature range, and in performance and
applicability in their timescale of experiments, particle lifetime, and wall losses. Typically, Teflon® materials such as fluorinated
ethylene polypropylene (FEP) are used for the chamber walls due to their inert and ultraviolet (UV) transmission properties.
However, environmental chambers have also been constructed of stainless steel (De Haan et al., 1999; Glowacki et al., 2007;
Duplissy et al., 2010; Wang et al., 2011; Massabò et al., 2018), aluminum (Saathoff et al., 2003), quartz (Barnes et al., 1994), and
Pyrex glass (Doussin et al., 1997). Some chambers are designed to be operated indoors (Doussin et al., 1997; Cocker et al., 2001;
Saathoff et al., 2003; Carter et al., 2005; Paulsen et al., 2005; Presto et al., 2005; King et al., 2009; Wang et al., 2011; Hu et al.,
2014; Wang et al., 2014); others are developed to be used outdoors (Jeffries et al., 1976; Becker, 1996; Klotz et al., 1998; Lee et
al., 2004; Rohrer et al., 2005; Chung et al., 2008; Im et al., 2014; Ren et al., 2017). A few chamber systems have been designed to
be portable (Shibuya et al., 1981; Hennigan et al., 2011; Bonn et al., 2013; Platt et al., 2013; Kaltsonoudis et al., 2019; Vu et al.,
2019). Several comprehensive reviews of existing environmental chambers have been published (Becker, 2006; Lee et al., 2009;
Seakins, 2010; Hidy, 2019).

In this paper, the development and characterization of the Captive Aerosol Growth and Evolution (CAGE) chamber system will
be discussed. The CAGE chambers are portable and designed to be used in the field. Observed changes in aerosols that are
generated and injected into the chambers provide useful information about atmospheric processing and the rates and mechanisms
of particle growth. Experience with a previous generation of this chamber system informed the design of that described here. That
previous version, also referred to as the quasi-atmospheric aerosol evolution study (QUALITY) chamber, consisted of a portable



1.2 m³ UV-transmitting upper reaction chamber with a sheet of gas-permeable membrane across the bottom that allowed ambient gas-phase species to enter the reaction chamber where seed particles were injected. A photo is shown in Fig. S1 in the Supplement.

They were used to study aging of primary particles, such as of black carbon in the polluted urban areas of Houston, TX, U.S. and Beijing, China (Glen, 2010; Peng et al., 2016, 2017). The design of the version described here differs considerably from the first generation and is therefore discussed in detail.

## 2 Design of CAGE chamber system

### 2.1 CAGE chambers

A sketch and photo of one of the two identical CAGE chambers that were constructed are shown in Fig. 1. The core of each chamber is a 1 m³ all-Teflon® cylindrical reactor constructed primarily of UV-transmitting 2 mil (0.05 mm) thick FEP film (a. in Fig. 1). The only non-FEP section is the ~0.2 mil (0.005 mm) gas-permeable expanded polytetrafluoroethylene (ePTFE; Phillips Scientific) membrane sheet (b.), located at the back end of the cylinder as shown in both the sketch and photo. The stainless steel internal support structure of the chamber is fully wrapped with highly reflective high-density PTFE thread tape that was first baked

at about 120 °C overnight to eliminate any residual volatile species. Each chamber is suspended in a stainless steel rectangular enclosure that was powder coated with a reflective white fluoroethylene vinyl ether (FEVE) fluoropolymer paint. The rectangular enclosure is covered by 4.8 mm thick UV-transmitting Plexiglas G-UVT acrylic sides (c.) in order to block wind that would otherwise increase mixing and, consequently, particle loss rate. FEP sheets cover all interior surfaces of the enclosure within about 0.5 m of the far (right), gas-permeable, end as shown in the sketch and photo. The FEP walls of the chamber bag are pulled tight

across internal rings so that a solid cylindrical shape can be maintained throughout experiments, minimizing turbulent mixing inside and the increased particle loss that would result.

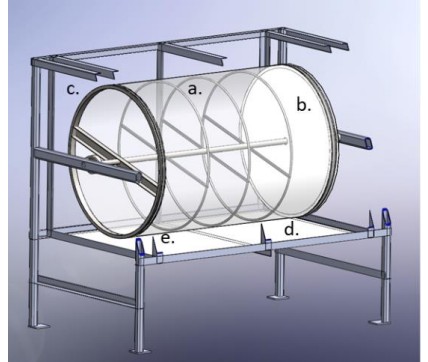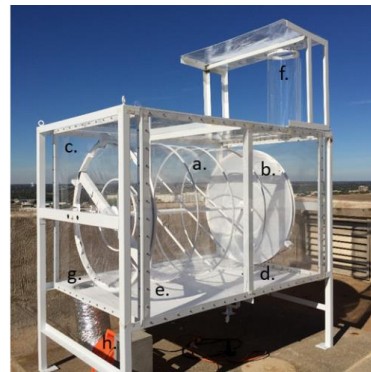

**Figure 1. Sketch and photo of a CAGE chamber.**


### 2.2 Ambient light source

Whereas most environmental chambers are illuminated by UV-emitting black lights, these utilize solar radiation for their light source. The overall light transmittance through the chamber and acrylic sheets is evident in the close-up photo in Fig. S2. The small loss of UV (and visible) solar radiation through the acrylic and FEP is partially offset by reflection off a highly UV-reflective

3.2 mm thick PTFE gasket sheet (Inertech SQ-S) just below the chamber (d.). At the site at which the field study described below was conducted, both chambers were oriented on the south side of an instrumented trailer and with their ePTFE membranes facing



north to minimize shading throughout the day. Prior to that study, an Ocean Insight Flame spectrometer was used to measure cosine-weighted solar spectral intensity outside and at a point between the bottom of one of the chambers and the reflective PTFE gasket (point e. in Fig. 1). The results shown in Fig. 2 represent the sum of the upwelling and downwelling measurements

(spectroradiometer receptor pointing straight down and up, respectively). The broad spectral transmittance of the chamber and enclosure sides and the broad spectral reflectance of the PTFE gasket result in the close chamber/outside match over the full UV range. Because the spectroradiometer receptor could not be positioned inside the chamber, the offset between the two curves is only an approximate indicator of the absolute agreement between the intensities inside and outside of the chamber.

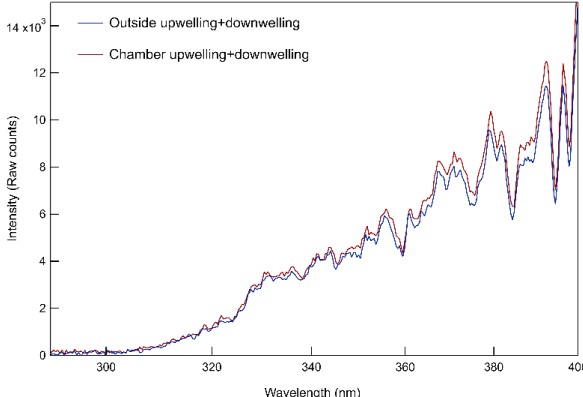

**Figure 2. Comparison of spectral intensity measured just below one of the chambers (around point e. in Fig. 1) and just outside of the chamber enclosure on a sunny day.**

### 2.3 Rotating chamber

The cylindrical reaction chamber is supported at both ends by mounted bearings that are attached to the enclosure frame. A DC-

powered motor connected to the chamber through a chain and pair of sprockets rotates it along its horizontal axis at approximately 1 revolution per minute (rpm). The slow rotation minimizes loss of large particles due to gravitational settling and losses of all sizes of particles by minimizing temperature gradients and dampening convective eddies. The technique of using a rotating chamber, or drum, to suspend sub-10 µm biological particles for extended periods of time is commonly employed in the field of aerobiology (Asgharian and Moss, 1992; Santarpia et al., 2019). The optimum rotation rate for particles smaller than 10 µm has

been shown to be around 1-2 rpm (Goldberg et al., 1958; Goldberg, 1971; Krumins et al., 2008), though experimental studies suggest higher rotation rates may be preferable (Sutton, 2005). Stainless steel tubes with 0.95 cm outer diameter that are used as aerosol injection and sampling ports extend out from the center axle on both ends of the chamber and are sealed using radial O-rings. Those tubes terminate 0.4 m inside both ends of the chamber at a radial distance of 8 cm out from the center of the axle.

### 2.4 Exchange of ambient air into the reaction chamber

Several $m^3$ $min^{-1}$ of ambient air is drawn through an FEP-lined inlet on top of the chamber (f. in the photo in Fig. 1) that is protected by an FEP-wrapped rain cover. The ambient air circulates behind the gas-permeable ePTFE membrane and then around the chamber to the opposite end of the enclosure where it is exhausted through a port (g. in the photo in Fig. 1) connected to a blower (h.; Allegro 9533) that is located below the acrylic frame. The FEP sheets covering the internal surfaces on the inlet end of the enclosure minimize contact of the air with any non-Teflon® surface prior to reaching the 0.9-$m^2$ ePTFE membrane. As is described

in Sect. 4, an effective exchange flow rate across the ePTFE membrane is estimated to be 33 L $min^{-1}$. The driving force of gas





exchange across the membrane and into the chamber is the difference in concentrations of gas-phase species between the chamber air and the ambient air that is flushed through the enclosure. Efficient gas exchange across the membrane maintains near-ambient trace gas concentrations in the chamber without diluting the captive aerosol, as would occur if ambient air were instead continuously pumped into the chamber, as is further discussed in Sect. 5. The membrane is similar to material commonly used in

filters used to collect aerosol samples and minimizes infiltration of ambient particles into the chamber, where they would mix with the narrow size mode populations of particles that are tracked over time.

### 2.5 Experimental procedure: Instrumentation

Similar to the dual-chamber systems described by Tkacik et al. (2017) and Kaltsonoudis et al. (2019), two identical chambers (called A and B) were utilized to evaluate the influence of differing conditions on the behavior of captive particles. For the

experiments described here, unperturbed ambient air was circulated behind each of the permeable membranes and the contrast in conditions was achieved by covering Chamber B with a light shield that reduced daytime UV intensity to below 1% of that in Chamber A. With the exception of the results from the chamber-ambient characterization experiment described below, only measurements from Chamber A will be described here.

Monodisperse seed particles were generated by atomizing an ammonium sulfate solution with a TSI 3076 atomizer, drying with a silica gel diffusion dryer, and separating a narrow size range with a differential mobility analyzer (DMA). The monodisperse particle mode was then injected into one of the chambers at a time, as discussed in Sect. 5. The instrumentation was configured to sample from both the inside of each of the two chambers and ambient air. Particle size distributions spanning the diameter range from 0.013 to 0.60 μm were measured using a custom-built scanning mobility particle sizer (SMPS) equipped with a TSI 3762

condensation particle counter (CPC) and a high flow DMA (Stolzenburg et al., 1998). The sampled aerosol was dried with a Nafion tube bundle and charge neutralized with a soft x-ray neutralizer prior to entering the DMA. A TSI UV-aerodynamic particle sizer (APS; 3314) was used in parallel with the SMPS to measure the aerodynamic size distributions of supermicron bioaerosol particles that were intermittently injected into the chambers.

Throughout roughly the first half of the study described here, the Mobile Air Quality Lab (MAQL) developed and operated by researchers from the University of Houston, Baylor University, and Rice University was located adjacent to the CAGE chambers at the field site. Instrumentation inside the MAQL measured trace gas concentrations and aerosol composition (Leong et al., 2017; Wallace et al., 2018). Ozone ($O_3$) was measured with a Thermo Environmental 49C analyzer. Nitric oxide (NO), nitrogen dioxide ($NO_2$), and the sum of nitrogen oxides ($NO_y$) were measured with an Air Quality Design, Inc (AQD) high sensitivity

chemiluminescence NO detector. $NO_y$ was measured by conversion to NO using a molybdenum oxide catalytic converter maintained at 320 °C. $NO_2$ was measured by photolytic conversion of $NO_2$ to NO using an AQD Blue Light Converter. An Ionicon Q300 quadrupole proton transfer reaction mass spectrometer (PTR-MS) measured concentrations of a fixed set of VOCs. Of the 19 VOCs measured, analysis here focused on the subset of species that were at least sometimes present at concentrations above their detection limit and were not significantly impacted by interference from other species detected at the same masses. The

species analyzed include acetaldehyde (detected mass = 45 Da), acetone (59), isoprene (69), methyl vinyl ketone+methacrolein (MVK+MACR; 71), methyl ethyl ketone (MEK; 73), benzene (79), toluene (93), and monoterpenes (137). A 2:1 α-pinene:β-pinene split in ambient air was assumed for the monoterpenes, partly based on regional emissions estimates with the Model of Emissions of Gases and Aerosols from Nature (MEGAN) (Guenther et al., 2012). Because the chamber inlet and outlet were not configured to allow simultaneous sampling of particles and trace gases, the gas analyzers sampled from the chambers only during a 3-day


period described below. Non-refractory submicron aerosol composition and mass concentration were measured with an Aerodyne high-resolution time-of-flight aerosol mass spectrometer (HR-ToF-AMS). Unlike the gas-phase instruments, the AMS intermittently sampled air from the chambers throughout the month the MAQL was operated at the field site. The approximate placement of the CAGE chambers and the valve and tubing configuration used to alternate between Chamber A and Chamber B for aerosol injection and sampling is depicted in Fig. 3. LabVIEW software was used to automate control of injection and sampling

systems, as well as to monitor parameters throughout the system.

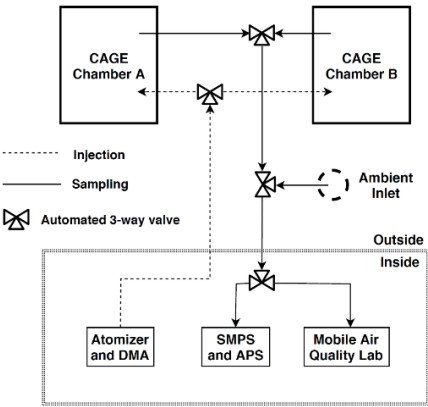

**Figure 3. Placement and orientation of the CAGE chambers relative to the instrument trailer, and tubing and valve configuration used to inject particles into and sample particles from both CAGE chambers.**

**3 Field site description**

The chambers were evaluated during a field study at the WG Jones State Forest (JSF) from August 15, 2016 to October 14, 2016. The JSF site is a roughly 2,000-acre (8 km$^2$) pine-dominated forest located between Conroe and The Woodlands in southeast Texas, U.S. The clearing in which the chambers were located, its location within the nearly rectangular state forest, and its proximity to the Houston area are shown in the set of satellite images in Fig. 4. A photo of the MAQL instrumentation, CAGE

chambers, and research trailers at the JSF sampling site is shown in Fig. S3.





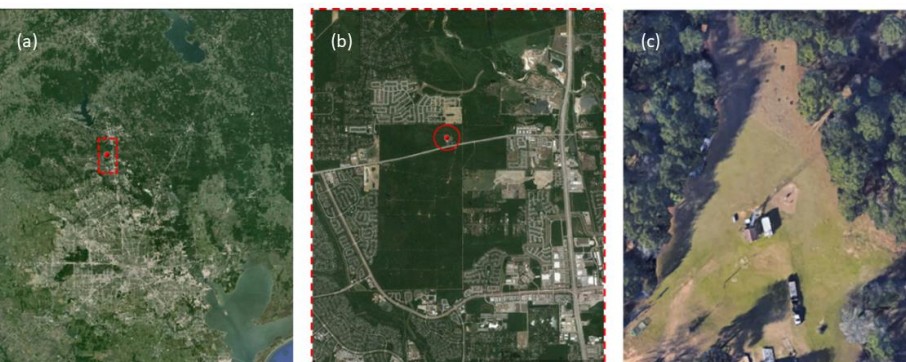

**Figure 4. Satellite images of the WG Jones State Forest (JSF) site at which the field study was conducted. The location of JSF relative to Houston is shown in (a), the location of the field site within the nearly rectangular JSF in (b), and the clearing at which the chambers and instrument trailers were located in (c). Map data © 2017 Google.**

The wind rose shown in Fig. S4 was calculated from the winds observed at the nearby Conroe, TX airport during the field study period. Those data highlight the prevalence of southeasterly winds, which bring the complex and concentrated mixture of pollutants from Houston into an area with high emissions of highly reactive biogenic hydrocarbons such as isoprene and monoterpenes. The goal of the field study was to investigate how fast and why particles grew in an environment that is impacted by high emissions rates of both anthropogenic and biogenic gases.

## 4 Relationship between ambient and chamber gas-phase composition

The measurements of trace gas concentrations made by instrumentation in the MAQL allowed the relationship between chamber and ambient air to be characterized by temporarily reconfiguring the mobile laboratory inlet to alternate between sampling from the chambers and from outside. These chamber-ambient characterization experiments were conducted over a 3-day break from the routine particle growth measurements, from midday 9/9/2016 to midday 9/12/2016. During these experiments, automated valves were controlled to produce the repeated sampling loop: Ambient (15 min) → Chamber A (15 min) → Ambient (15 min) → Chamber B (15 min). Unlike the rest of the 2-month study, Chamber B was uncovered for these experiments in order to assess the chamber-to-chamber consistency. Similar to what has been observed during prior chamber-ambient comparisons, measured trace gas concentrations in each chamber could be explained by treating the volume as a continuous stirred-tank reactor (CSTR). The resulting rate of change of the concentration of any of the trace gases can then be expressed as:

$$\frac{dC_{ch}}{dt} = P - L + \frac{Q_{ex}}{V_{ch}}C_{amb} - \frac{Q_{ex}}{V_{ch}}C_{ch} \tag{1}$$

where $C_{ch}$ is the concentration in the chamber, $C_{amb}$ is the ambient concentration, $V_{ch}$ is the volume of the chamber ($\approx$1000 L), P and L are the per unit volume rates of chemical production and loss in the chamber, respectively, and $Q_{ex}$ is an effective exchange flow rate across the ePTFE membrane. The $Q_{ex}$ cannot be measured directly and is instead estimated using Eq. (1) and time series of concentrations measured in the chamber and outside. It is best determined for a gas that has negligible chemical production and loss (P = L = 0) and that is present at concentrations well above the instrument detection limit. $NO_y$ best satisfied those requirements among the species measured during the characterization experiments. The value of $Q_{ex}$ was estimated as that resulting in the





maximum correlation ($r^2$) between the time series of the concentration measured in the chambers and that calculated from the ambient time series using Eq. (1). A $Q_{ex}$ of 33 L min$^{-1}$ resulted in a peak $r^2$ of about 0.97 for both chambers (Fig. S5). The resulting 3-day time series for $NO_y$ in Fig. 5 shows that the mixing ratios in the chambers closely match those calculated from the ambient measurements. Treating the chamber as a CSTR captures the observed smoothing of short duration peaks and troughs in the

ambient data. Fig. 6 presents the same data for $NO_y$ (and other species as discussed below) as pairs of mixing ratios i) measured in the chambers (y-axis) and ii) calculated from the ambient measurements (x-axis). The best fit lines through the $NO_y$ pairs have slopes of 0.98 and 1.00 for Chambers A and B, respectively. For all other measured species (or groups of species), chemical loss and/or production over the approximate $V_{ch}/Q_{ex}$ = 30 min residence in the chambers may be significant. For free radicals and other highly reactive or condensable species with typical atmospheric lifetimes much shorter than 30 min (e.g., hydroxyl radical, OH·,

and nitrate radical, NO$_3$·), exchange across the ePTFE membrane is insignificant and P ≈ L in Eq. (1).

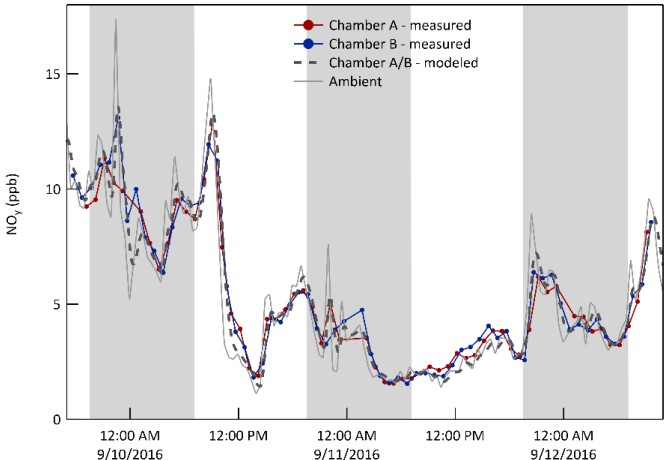

**Figure 5. Time series of $NO_y$ mixing ratio (ppb) measured in both chambers, measured just outside of them, and calculated from the ambient time series by modeling them as CSTRs with an exchange flow rate, $Q_{ex}$, of 33 L min$^{-1}$. The shaded bands represent nighttime.**


A simple CSTR-zero-dimensional (0D) model was developed to interpret the results from the chamber-ambient characterization experiment and to subsequently use ambient measurements made throughout the period when the MAQL was at the site to estimate concentrations of measured and unmeasured species in the chambers. The model numerically integrates the time-dependent changes resulting from the 33 L min$^{-1}$ gas exchange and the reactions listed in Table S1. Only those reactions expected to have

significant influence on the concentrations of measured or otherwise important species were included. Most photolysis rate constants were calculated from measured spectral intensity. The only exception is that for $NO_2$ during daytime, which was instead calculated assuming ambient NO, $NO_2$, and $O_3$ concentrations satisfied the photostationary state relationship, $J_{NO_2} = k[O_3][NO]/[NO_2]$, where [ ] indicates concentration. Some reactions are intentionally not balanced where one or more of the products are not tracked in the model (e.g., NO$_3$· + RO$_2$· → $NO_2$) and some reactions combine a series of steps wherein only the

step controlling the overall rate is included (e.g., $NO_2$ + hν → NO + $O_3$). The 0D model was used to calculate concentrations of measured (e.g., $O_3$ and isoprene) and unmeasured (e.g., OH· and NO$_3$·) gases and to estimate secondary aerosol production rates resulting from reactions of $SO_2$ with OH· and reactions of hydrocarbons with OH·, NO$_3$·, and $O_3$. The concentrations measured in the chambers for almost all species agree well with those calculated from the ambient measurements, as is reflected in the near 1.0 best fit slopes and high $r^2$ values shown in Fig. 6. There is more scatter about the 1:1 line for the monoterpenes than for other





species, which is largely the result of noise in the measurements, as the mixing ratios were close to the detection limit for most of the measurement period. Uncertainty in the relative abundance of the different monoterpene species may also contribute to this noise because reaction rates vary considerably among the species, while only the sum was measured by the PTR-MS. Measured concentrations of MEK were generally higher than predicted, which is believed to be the result of production in the chamber from oxidation of species such as butane that were not measured and therefore not included in the model. Described below are measured and modeled concentration time series for the $NO/NO_2/O_3$, isoprene/MVK+MACR, and $O_3$/acetaldehyde systems that, collectively, elucidate the relationship between conditions in the chamber and those outside.

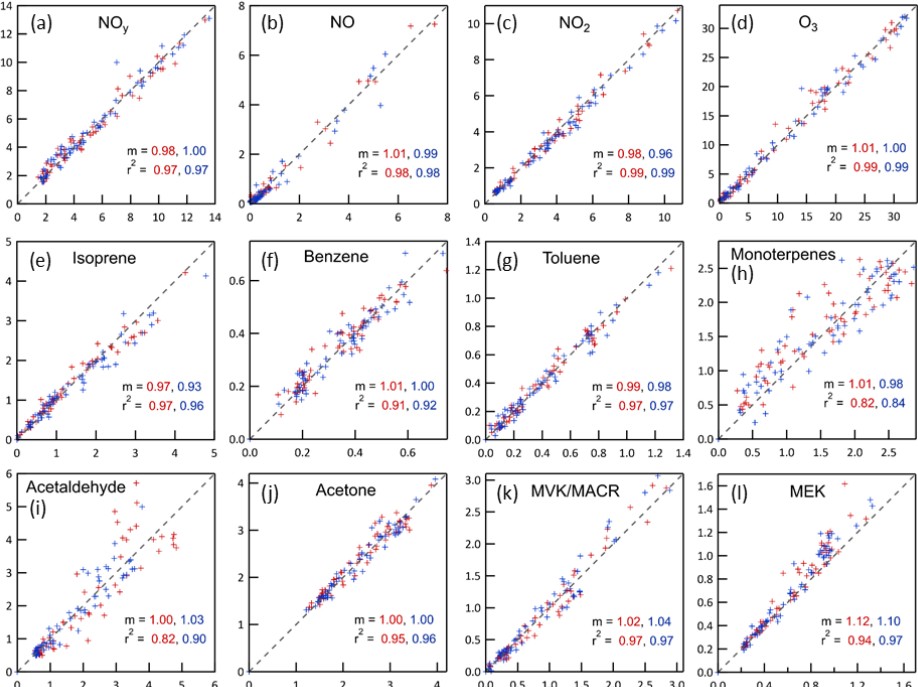

**Figure 6. Relationships between mixing ratios (ppb) expected in the chambers calculated from the ambient time series using the CSTR-0D model (x-axes) and measured in the chambers (y-axes). For all graphs, the red markers and text are for Chamber A and the blue are for Chamber B. The dashed lines shown in all graphs are 1:1 lines. The slopes, _m_, are for best fit lines forced through the origin. MVK/MACR = methyl vinyl ketone/methacrolein, both of which are measured at the same mass by the PTR-MS. MEK = methyl ethyl ketone.**

$NO/NO_2/O_3$: Whereas the concentration of the sum of all nitrogen oxides ($NO_y$) is roughly conserved over the chamber-ambient mixing time, that of its more reactive components may not be. During the daytime, approximately steady state cycling between NO, $NO_2$, and $O_3$ minimizes any differences between chamber and ambient concentrations. At night, however, reaction of $O_3$ with both NO and $NO_2$ results in concentrations in the chamber that are, conceptually, what would be expected about 30 min (= $V_{ch}/Q_{ex}$) downwind of its physical location. The model captures the influence of the reactions, resulting in close agreement between the expected and measured mixing ratios for both NO and $NO_2$, as is shown in the time series in Fig. 7 and in the clustering of points around the 1:1 lines in Fig. 6. For clarity, only the results for one of the two chambers (B) are shown in the time series, while the measurement/model pairs from both chambers are shown in Fig. 6.



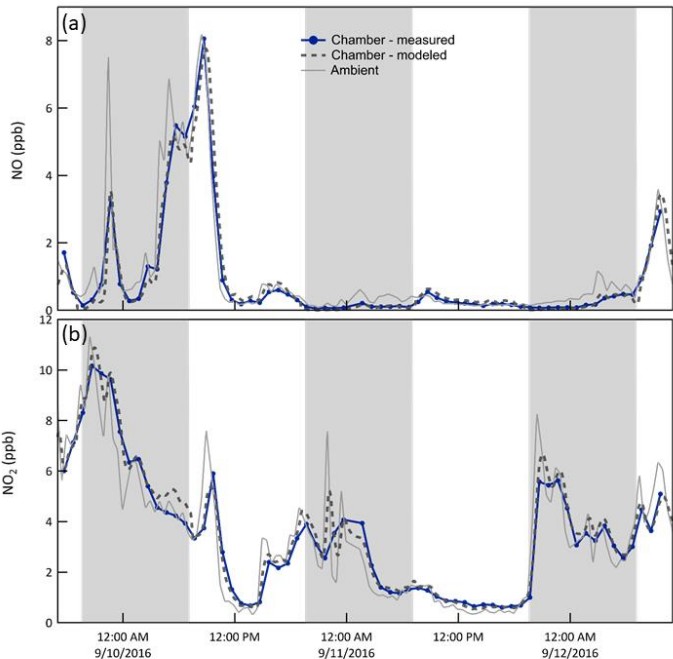

**Figure 7. Time series of NO and NO₂ mixing ratios i) measured in chamber B (solid blue), ii) expected in the chamber as calculated from the CSTR-box model (dashed), and iii) measured outside (solid gray). Smoothing of the spikes in the ambient time series results from treatment of the chamber as a CSTR. The shaded bands represent nighttime.**

Isoprene/MVK+MACR: Oxidation of reactive hydrocarbons by OH·, $O_3$, and $NO_3$· creates a mixture of products that may subsequently react or may condense on the particles that were injected into or formed in the chamber. Biogenic VOCs including isoprene and monoterpenes were typically the most concentrated hydrocarbons at the forested site. Isoprene chemistry is most important during the daytime as its emission rate is largely controlled by solar intensity, whereas the temperature-dependent emission of monoterpenes varies comparatively little throughout the day/night. Fig. 8 shows the influence of in-chamber chemistry on the mixing ratios of isoprene and its oxidation products MVK+MACR (only the sum of the two was measured with the PTR-MS). During the daytime and early evening when concentrations of OH· and $O_3/NO_3$·, respectively, are high, both the expected and measured mixing ratios of isoprene are lower and those of MVK+MACR are higher than those measured outside. For both species the CSTR-0D model captures the features in the time series quite well with resulting average best fit slopes and $r^2$ of 0.95 and 0.97 for isoprene and 1.03 and 0.97 for MVK+MACR.





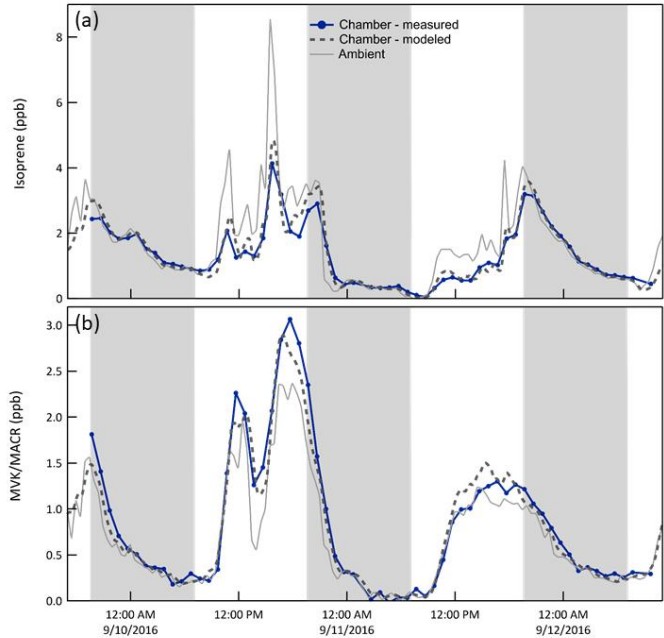

**Figure 8. Mixing ratio time series of isoprene and its reaction products methyl vinyl ketone (MVK) + methacrolein (MACR) i) measured and ii) expected in Chamber B, and iii) measured outside. Isoprene that enters through the ePTFE membrane reacts with OH·, O₃, and NO₃· in the chamber, resulting in a lower mixing ratio than outside. The shaded bands represent nighttime.**

O₃/acetaldehyde: Reaction of $O_3$ with the Teflon® walls and/or impurities on those walls results in a slightly lower concentration in the chamber than outside, as shown in Fig. 9a. To represent this in the model, an $O_3$ + Wall reaction was included and its rate constant adjusted to match the observations in each chamber. Additionally, as reported elsewhere (De Gouw and Warneke, 2007), surface reaction of $O_3$ produces acetaldehyde, which is believed to be responsible for the higher concentrations in the chamber than outside evident in Fig. 9b. Thus, an acetaldehyde yield from the $O_3$ + Wall reaction was used as an additional tuning parameter. With the corrections, the modeled $O_3$ matches that observed very well, with average best fit slope of 1.00 and $r^2$ of 0.99. The corresponding values for acetaldehyde are 1.02 and 0.86.





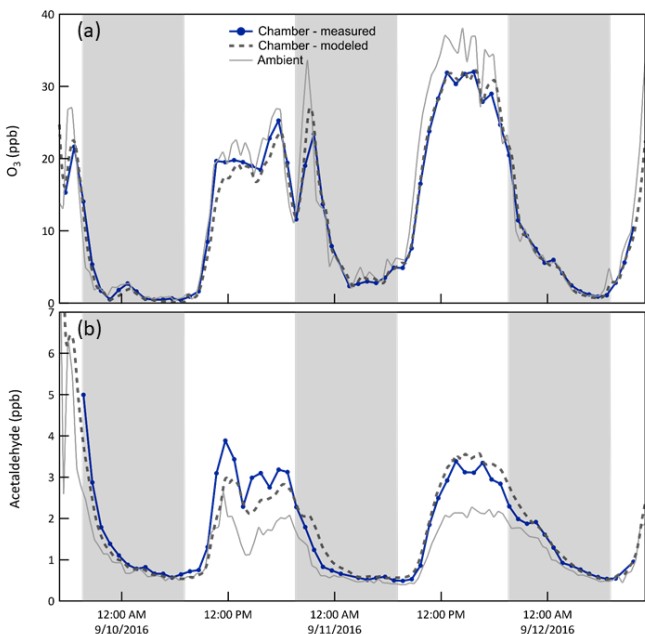

**Figure 9. Time series of $O_3$ (a) and acetaldehyde (b) mixing ratios i) measured and ii) expected in Chamber B, and iii) measured outside. Ozone mixing ratios were slightly lower in the chamber than outside, which is believed to be the result of reactions with the Teflon® surfaces or with impurities on those surfaces. The loss rate constant was adjusted to match the observations. The shaded bands represent nighttime.**


The other tuning parameters in the model were used to estimate OH· concentration. Specifically, an overall OH· + X loss rate (or OH reactivity) of 2 s$^{-1}$ (or $\tau = 0.5$ s) was assumed, as was a continuous source of nitrous acid, HONO, from the Teflon® surfaces (Rohrer et al., 2005). Photolysis of that HONO and of the modeled $O_3$ were assumed to be the only OH· sources. While the resulting

OH· concentration is not well constrained, the modeled influence of OH· on reactive species such as isoprene is consistent with that observed. Overall, the improved understanding of the CAGE chambers in general and the CSTR-box model in particular that came from the chamber-ambient comparison experiment increases the accuracy with which changes in captive particles can be connected with the responsible environmental conditions.

**5 Particle addition and sampling strategy**

To quantify particle growth rates and connect them with the responsible secondary aerosol formation and particle evaporation, sub-0.1-μm size-classified particles were repeatedly injected into the chambers throughout the two-month study. Ammonium sulfate was selected due to its common use as a seed aerosol in chamber studies and because it often represents a significant component of atmospheric aerosols. To detect and accurately quantify changes that are typically between -1 and 10 nm h$^{-1}$, the atomized aerosol was first size-selected with a DMA to generate a monodisperse population. Throughout the experiments, an

SMPS was used to measure the particle size distribution in each chamber twice per hour. The tracked mode size distributions were fitted using a lognormal function and the dry particle diameter, $D_p$, and number concentration, N, parameters of the fits were used to calculate diameter growth rate and concentration loss rate, respectively. A new monodisperse mode was added as soon as the previously injected mode became difficult to track. With this approach, growth rates were determined nearly 24 h day$^{-1}$.



Condensable species that are produced are expected to be distributed among the particle population in the chamber. The division among those particles can depend on properties of the condensable species and of the particles, though results from this study suggest a simple particle surface area dependence. Because of competition for the condensable species among all particles in the chamber, the growth rate of the tracked mode will be affected by the total surface area concentration or condensation sink. To minimize that influence, an additional monodisperse ammonium sulfate mode centered at 0.3 µm was maintained, with new

injections triggered automatically each time the surface area concentration calculated from the SMPS-measured size distribution fell below 40 µm$^2$ cm$^{-3}$. In addition to the injected ammonium sulfate particles, new particle formation events would sometimes occur inside the chambers just as they do in the atmosphere (Kulmala et al., 2004). The nucleation or NPF modes are broad relative to those injected, but are still typically narrow enough to track. The lower time series in Fig. 10 shows an example of the chamber size distributions measured by the SMPS over a 3.5-day period in September, 2016. The size distributions measured just outside

the chambers during the same period are shown in Fig 10a to highlight the relative difficulty in connecting time variation in ambient measurements to the rate and cause(s) of particle growth. The modes labeled "IM" are the injected monodisperse ammonium sulfate modes and those labeled "NM" are nucleation modes consisting of particles that formed and grew in the chamber. The repeated injections into the surface area mode (SAM) results in the roughly horizontal band centered at about 0.3 µm. To the right of the intensity graph is an x-y representation of the size distribution measured at the time indicated by the rectangular box in the

intensity graph. Also shown in the x-y graph is the particle surface area concentration size distribution, which illustrates the extent to which the total concentration can be controlled by the SAM and is minimally impacted by the smaller particle tracked mode.

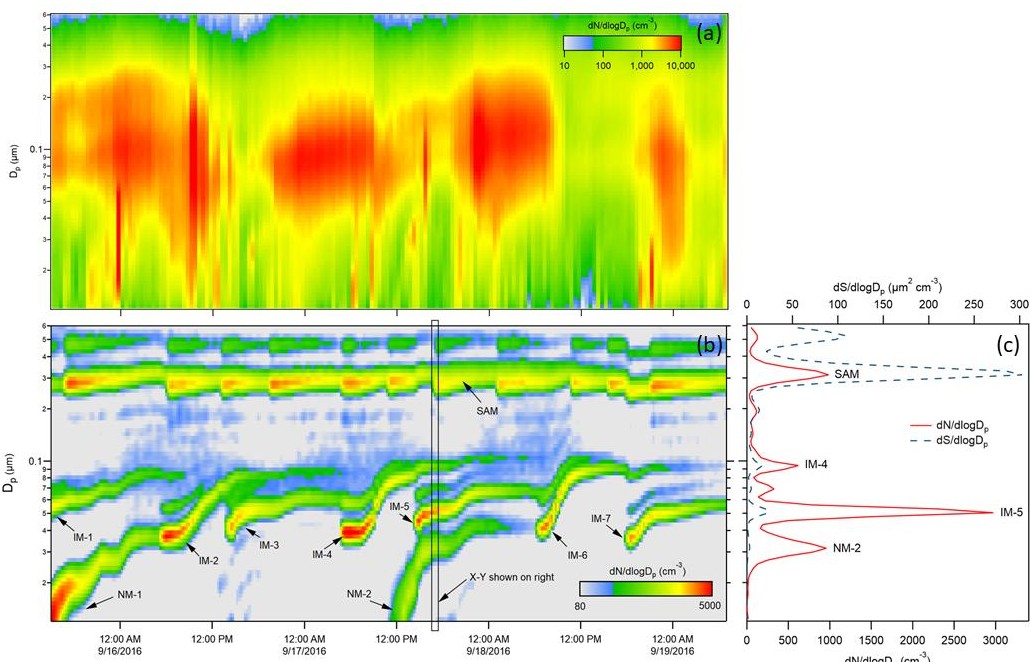

**Figure 10. (a): Ambient aerosol size distribution time series over 3.5 days during the 2016 study. (b): Chamber A size distribution time**
**series over the same period. IM = injected mode, NM = nucleation mode, SAM = surface area mode. (c): x-y presentation of the size distribution measured at the time indicated by the rectangle in the intensity graph. N = number concentration and S = surface area concentration.**

As noted above, the tracked modes were fitted with lognormals to extract the time-dependent mode diameters. The result for the

same 3.5-day period is shown in Fig. 11, where each of the curves represents one tracked mode. Different colors are used for



different modes that overlapped in time and shaded bands are included to indicate nighttime. Figure 11a shows the integrated surface area concentration during the same period, with the saw tooth pattern resulting from the automatic injections of SAM particles each time the integrated surface area concentration fell below 40 $\mu m^2$ $cm^{-3}$. The x-y representation in Fig 11c shows the lognormal fits to the same nucleation mode and two injected modes identified in Fig. 10. The compilation of the growth curves of all of the modes tracked during the 2016 study is presented in Fig. S6.


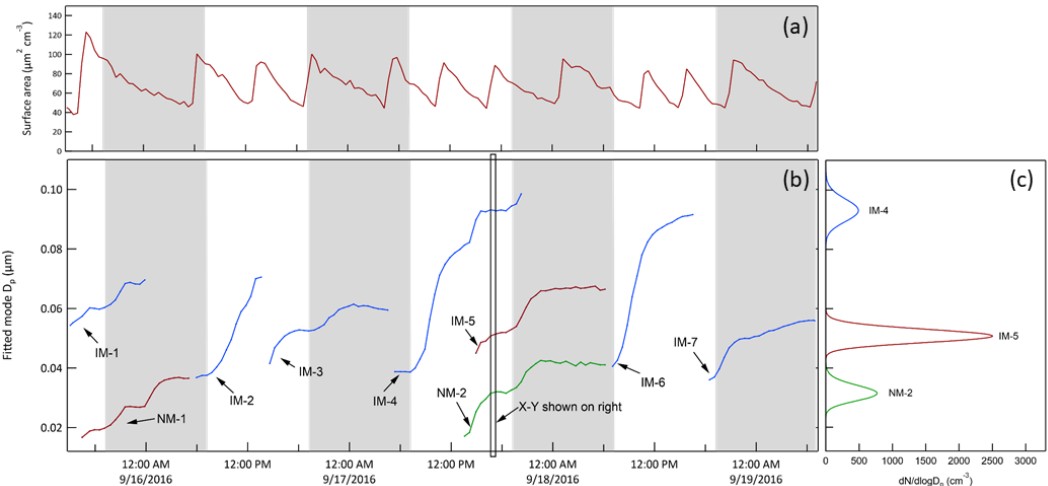

**Figure 11. (b): Time series of the lognormal fit diameters of injected and nucleation modes identified in Fig. 10. Different colors are used when two or modes overlap in time. (c): Lognormal fits to the three modes tracked at the time indicated by the rectangle in the time series. (a): Integrated surface area concentration showing the result of slow decay followed by an automatic injection of ~0.3 μm particles**
**each time the concentration fell below 40 μm² cm⁻³. The shaded bands represent nighttime.**

For each tracked mode the growth rate (GR) was calculated as the change in lognormal fit $D_p$ between two successive measurements, divided by the time difference between them, $GR = \Delta D_p / \Delta t$, as presented in Fig. 12 for the same 3.5-day example period. The integrated surface area concentration time series shown in Fig. 11 is also included in Fig. 12. The vertical dashed lines 365 in Fig. 12 correspond to the times at which surface area mode particles were injected. As is true for the full two-month dataset, there is no obvious reflection of the surface area concentration pattern in the tracked mode growth rate curves, suggesting the range was sufficiently narrow and justifying the decision to use the calculated growth rates without any correction. An important feature in Fig. 12 is the similarity in GR among multiple modes that were tracked simultaneously. This lack of size dependence suggests that condensational growth resulted in a rate of change of the volume of the particles that was proportional to their surface area. It 370 simplifies use of the data as a diameter-dependent correction would otherwise be required. Figure 13 shows the comparison of all growth rates calculated from simultaneously tracked modes. The general clustering around the 1:1 line and the lack of a strong size dependence supports the interpretation that the growth rate is independent of particle size.



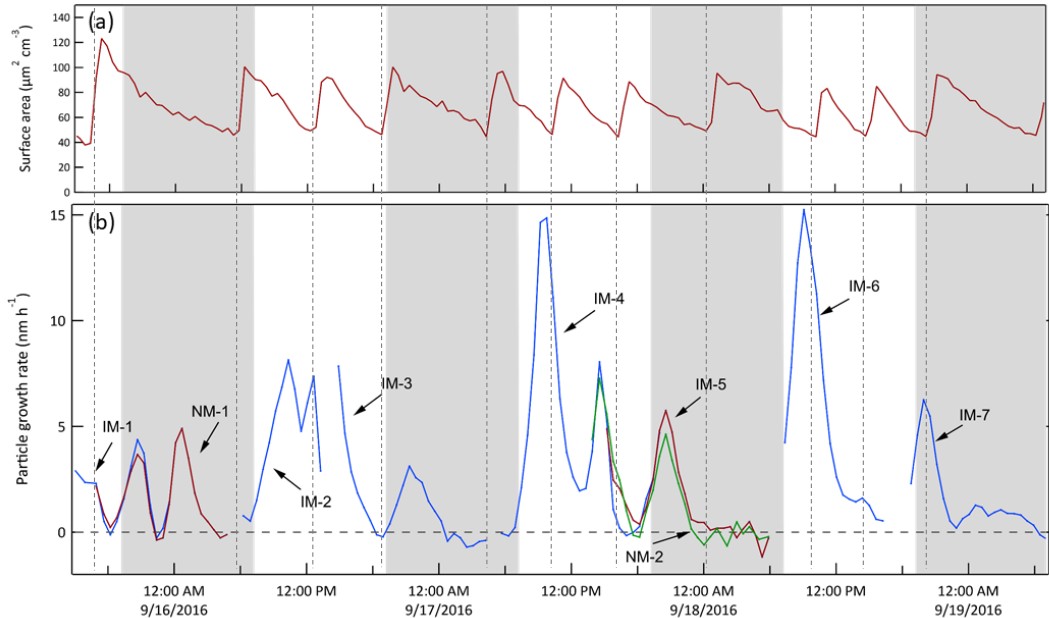

**Figure 12. Particle growth rates calculated from the time series of lognormal fit diameters shown in Fig. 11. The different colors correspond to those used in Fig. 11. The integrated surface area concentration time series shown in Fig. 11 is included here to note the absence of any obvious reflection of that pattern in the growth rates. The shaded bands represent nighttime.**

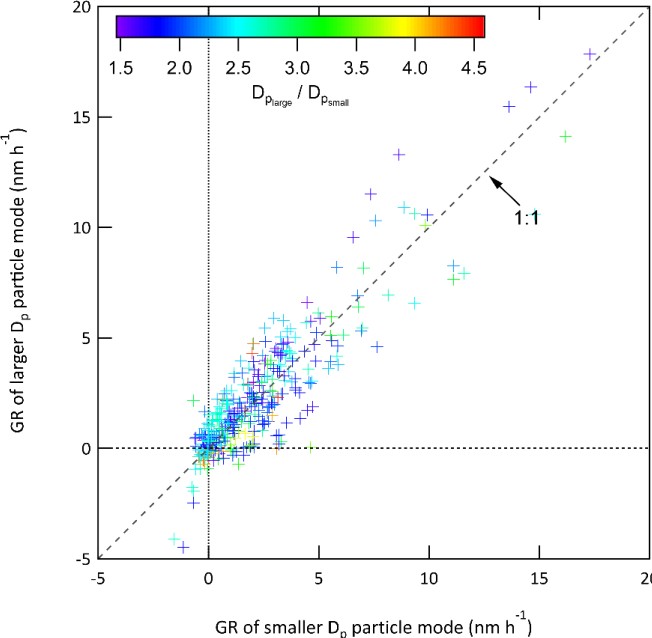

**Figure 13. Comparison of all pairs of growth rates calculated for multiple modes tracked at the same time. The GR of the smaller diameter mode is plotted on the x-axis and that of the larger diameter mode on the y-axis. The ratio of the diameter of the lognormal fit of the larger particle mode to that of the smaller particle mode is indicated by marker color.**





The CAGE chambers were designed to permit experiments on captive aerosols for periods ranging from hours to more than a day. For the approach employed with some chambers of continuously adding and extracting equal flow rates, the particle lifetime would

be too short if flow rates comparable to $Q_{ex}$ were used ($\tau \sim$ 30 min), while the gas-phase composition would differ too much from that outside if a much lower flow rate was used. Thus, by only exchanging the gases and not the particles across the gas-permeable membrane it is possible to conduct long-duration experiments under ambient conditions. Particle retention was further increased by rotating the chambers and by taking steps to minimize static charge on the Teflon® surfaces. Figure 14 summarizes the resulting distribution of lifetimes measured for different particle size populations, and identifies the techniques and instruments used to

quantify them. The three histograms represent particle lifetime distributions for distinct particle populations. As expected, loss rates are lowest and lifetime highest for the 0.3 μm particles that are in the size range at which the combined influence of diffusional and settling losses reaches a minimum. The average lifetime of bioaerosol particles with an average $D_p$ of 2.4 μm was 3.9 h. Even neglecting losses due to sample flow extraction and electrostatic attraction, a lifetime of less than 1 h would be expected for a non-rotating chamber with the same 0.53 m radius and the same 2.4 μm particles, which have a settling velocity of about 0.65 m h$^{-1}$.

Loss rates were typically highest during the daytime as solar heating promoted convection in the chambers.

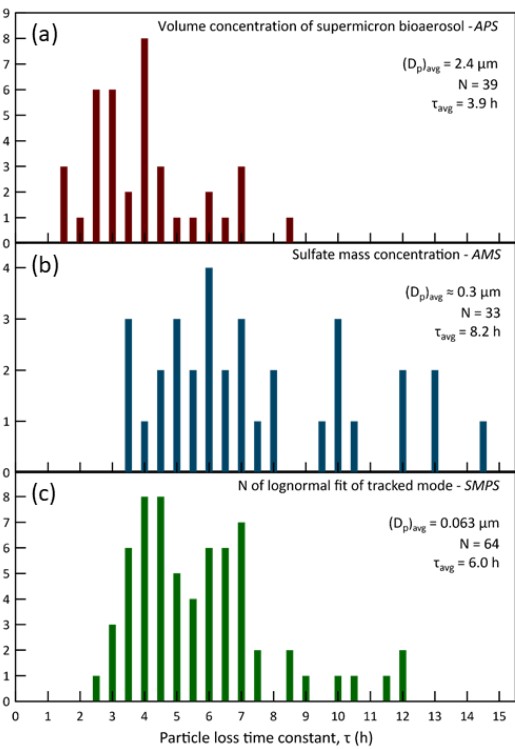

**Figure 14. Exponential time constants for particle loss in the chambers during the 2016 study. Graph (a) presents the distribution of**
**lifetimes of injected bioaerosol particles, which had an average diameter of 2.4 μm. The loss rates were determined from exponential decay fits to the supermicron volume concentration that was calculated from measurements by an APS. Graph (b) presents the distribution of lifetimes of 0.3 μm diameter ammonium sulfate particles intermittently injected to maintain a stable surface area concentration in the chambers. The loss rates were determined from exponential decay fits to the sulfate mass concentration measured with an HR-ToF-AMS. Graph (c) presents the distribution of lifetimes of "tracked mode" particles having a study-average diameter of**
**0.063 μm. The loss rates were determined from exponential decay fits to the number concentration parameter of lognormal fits to the narrow mode distributions measured with an SMPS. The y-axis values are the number of times the calculated loss rates fell within each 0.5 h bin.**


Simply taking the average of the lifetimes for the three particle populations gives a particle lifetime of 6.0 h. This is quite high for

such a small chamber with a correspondingly high surface area to volume ratio. In fact, it is higher than those reported for much larger chambers used to study secondary aerosol formation, as is presented in Table 1 that combines the data for CAGE with other loss rates summarized by Wang et al. (2014). A caveat of the simple comparison with other chambers is that for CAGE the lifetimes were averaged over daytime and nighttime conditions, whereas for at least some of the other chambers the values were determined during daytime or when artificial lights were on.


**Table 1. Comparison of chamber particle loss rates. Copied from Wang et al. (2014) with CAGE data added.**

| Chamber | Volume (m³) | Wall material | Wall loss rate (h⁻¹) | Particle lifetime (h) | Reference |
|---|---|---|---|---|---|
| CAGE | 1 | FEP | 0.17 | 6.0 | This work |
| GIG-CAC | 30 | FEP | 0.17 | 5.9 | (Wang et al., 2014) |
| PSI | 27 | FEP | 0.21 | 4.8 | (Paulsen et al., 2005) |
| Caltech | 28 | FEP | 0.20 | 5.0 | (Cocker et al., 2001) |
| UCR | 90 | FEP | 0.29 | 3.4 | (Carter et al., 2005) |
| EUPHORE | 200 | FEP | 0.18 | 5.6 | (Martin-Reviejo and Wirtz, 2005) |
| SAPHIR | 270 | FEP | 0.27 | 3.7 | (Rollins et al., 2009) |
| CMU | 12 | FEP | 0.40 | 2.5 | (Donahue et al., 2012) |

## 6 Results and discussion: Connecting gas-phase and aerosol-phase measurements with aerosol production estimates

Quantifying secondary aerosol formation in a chamber by diameter growth rate as is done here is atypical. Almost all chamber studies instead measure and report the change in particle volume or mass concentration over time. The result can conveniently be

related to secondary aerosol mass yields, which can then be used in atmospheric models that predict aerosol production following reaction of various precursors. However, traditional chamber experiments often use precursor concentrations much higher than observed at locations such as JSF, particularly at times corresponding to the growth rate minima commonly observed in the early morning and late afternoon. Evidence of the difficulty of tracking secondary aerosol mass production in ambient concentration chambers such as these is demonstrated in Fig. 15, which shows the GR time series from Fig. 12 together with the organic aerosol

production rate, $dM_{org}/dt$, calculated from HR-ToF-AMS measurements. Though production of inorganic sulfate and nitrate aerosol could also contribute to the total aerosol production rate, the organic component is expected to dominate at forested sites in general and, as is shown in Fig. S7, was found to contribute an average of about 74% of the ambient non-refractory submicron mass concentration during this study. The rate of change of organic aerosol mass concentration in the chambers was first corrected for losses due to flow extraction and wall deposition by normalizing with respect to the concurrently measured sulfate aerosol

concentration,

$$\frac{dM_{org}}{dt} = \frac{(M_{org})_{i+1} - (M_{org})_i \frac{(M_{SO_4})_{i+1}}{(M_{SO_4})_i}}{t_{i+1} - t_i}. \qquad (2)$$

Data were used only from those periods during which $M_{SO_4}$ exhibited the characteristic exponential decay expected as the initially

high concentration from an ammonium sulfate SAM injection falls due to a constant loss rate. Figure 15a shows those decay





profiles and corresponding loss time constants. Though averaging the results over longer time periods would reduce the noise, it would also reduce the utility of the data, as the time dependence would be obscured.

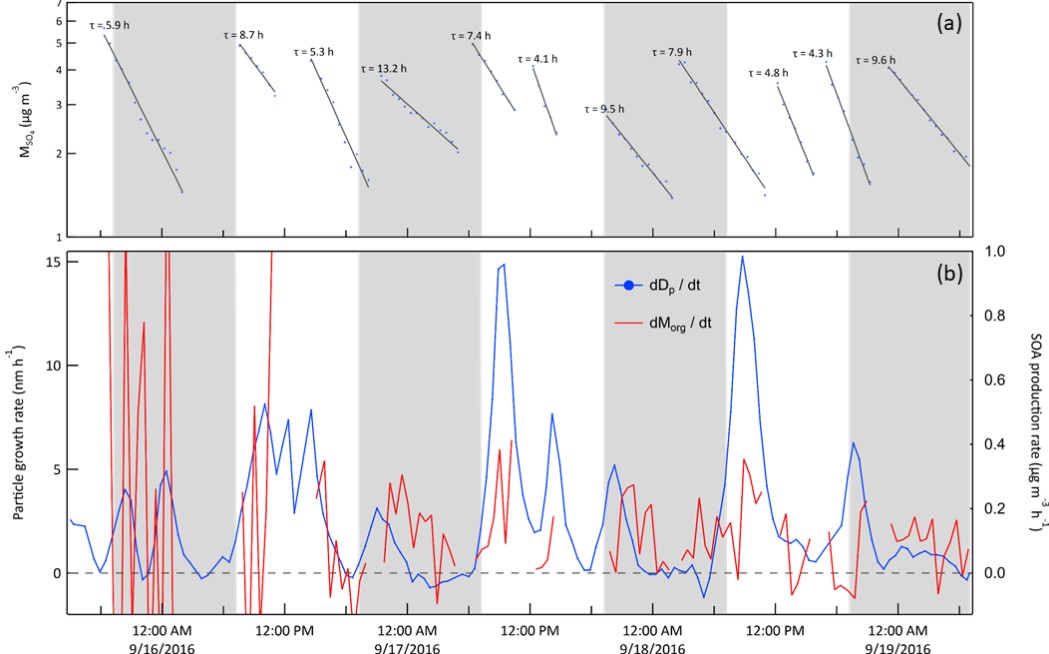

**Figure 15. (a): Sulfate mass concentration in Chamber A measured with the HR-ToF-AMS during the same 3.5-day example period as in Fig. 10, 11, and 12. The lines and time constants are for the exponential decay fits to the concentrations. (b): Averaged diameter growth rate from Fig. 12 (blue) and calculated rate of change of organic aerosol mass concentration (red). The rate of change of the organic mass concentration was corrected for loss using the sulfate mass using Eq. (2). The contrast between the two highlights the challenge in quantifying secondary aerosol production under ambient conditions using measurements of aerosol volume or mass concentration and**
**motivates instead tracking narrow size distribution modes. The shaded bands represent nighttime.**

The diel profile of growth rate averaged over the full study period is shown in Fig. 16. An initial observation of the overall pattern is that the average GR is positive for every hour of the day. If growth resulted primarily from equilibrium partitioning of semi-volatile organics, the GR would be positive as the concentration of those gas-phase species increased with time and negative when they decreased, with an average at a fixed location over a long enough period close to zero. The combination of the observation that GR is diameter-independent and that it is almost always positive suggests that the species responsible for much of the growth had very low volatility and irreversibly condensed on the particles at a rate controlled mostly by their surface area.





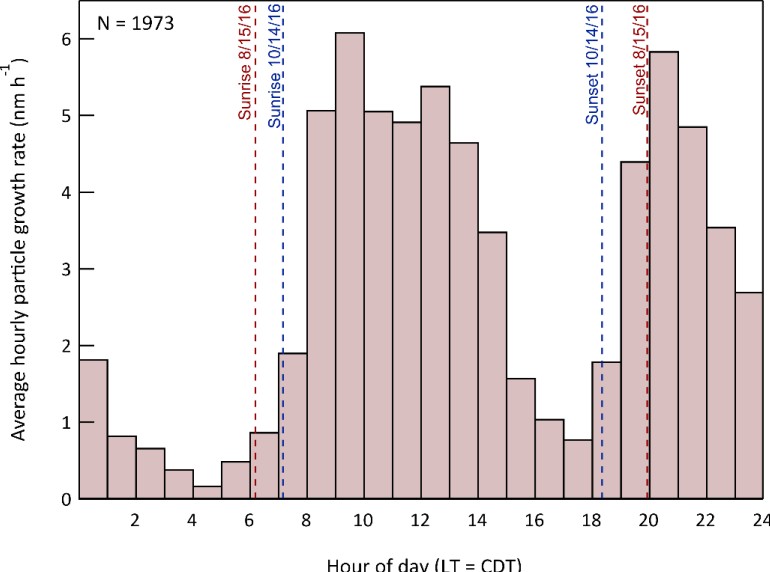

**Figure 16. Hourly average particle growth rate during the study. A total of 1,973 values were used to construct the histogram. The times of sunrise and sunset for the first (August 15) and last (October 14) day of measurements are also indicated.**

The bimodality evident in the histogram in Fig. 16 results from active OH·-driven aerosol production during the day and $NO_3$·- and $O_3$-driven production at night. Figure 17 shows the daily profiles of the mean and 25th/75th percentile range for the calculated in-chamber concentrations of the important oxidants (OH·, $O_3$, and $NO_3$·) and most important anthropogenic (toluene) and biogenic (isoprene and the monoterpenes) secondary organic aerosol precursors. The average GR profile from Fig. 16 is included in each of the columns of graphs in Fig. 17 to more clearly show the connection between the gas-phase concentrations and the resulting aerosol production. The similarity between the OH· and daytime GR profiles and between the $NO_3$· and nighttime GR profiles is apparent. The leftward shift in the daytime GR profile relative to that of OH· and the rightward shift in the nighttime GR profile relative to that of $NO_3$· is believed to result from variation in the hydrocarbon precursor concentrations, as the aerosol production rate is dependent on the product of the oxidant and precursor concentrations. Most importantly, the concentrations of both toluene and the monoterpenes decrease during the morning as the mixed layer deepens and then increase in the early evening as the pattern reverses and vertical mixing is limited.





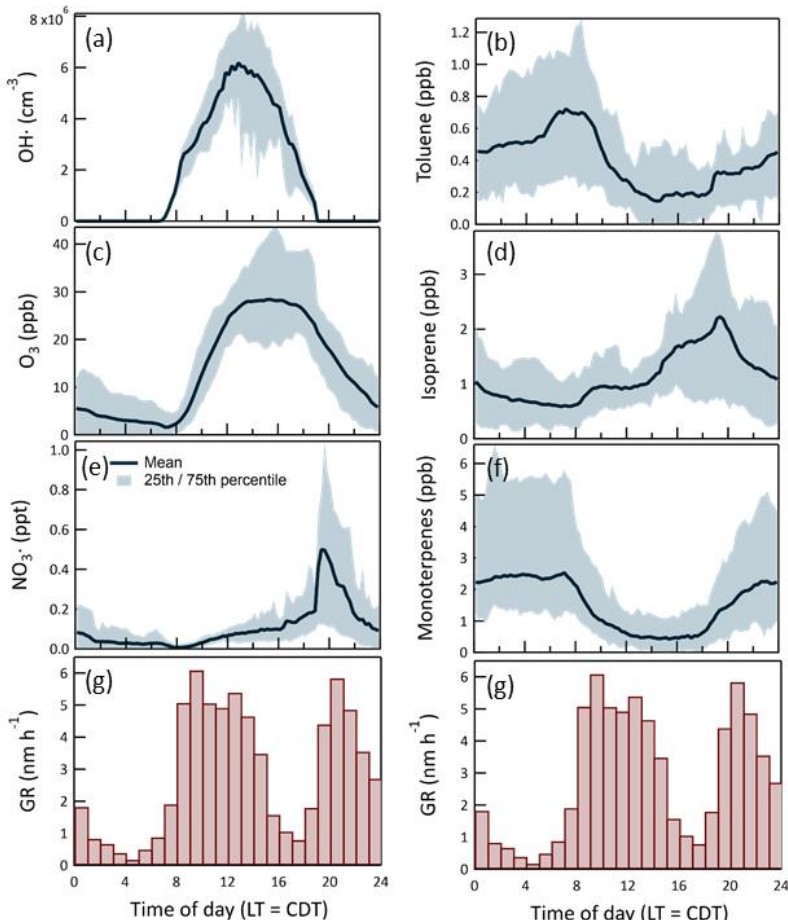

**Figure 17. Time of day-dependent mean and 25th/75th percentiles for the most important oxidants (a), (c), (e) and secondary organic aerosol precursors (b), (d), (f). Hourly averaged particle growth rates during the 2016 study are shown at the bottom (g). A total of 1,973 values were used to construct the histogram.**

Support for the broad interpretation of the relationship between the patterns in the GR and gas-phase concentrations in Fig. 17 comes from estimates of secondary aerosol production rate with the CSTR-0D model. The results are used to further interpret temporal variability in GR during the day/night and between days, and to assess the rates of aerosol production resulting from specific oxidant and precursor combinations. The estimated aerosol production at each time step is calculated as

$$P_{SA} = \frac{1}{N_A} \sum_i k_i [Precursor][Oxidant] MW_{precursor} Y_i' , \qquad (3)$$

where $P_{SA}$ is the secondary aerosol production rate, $N_A$ is Avogadro's number, $k_i[Precursor][Oxidant]$ is the reaction rate between a precursor and oxidant, MW is the molecular weight of the precursor, and $Y_i'$ is an effective aerosol production yield that was adjusted such that (arbitrarily) the value of $P_{SA}$ (in µg m$^{-3}$ h$^{-1}$) closely matched that of GR (in nm h$^{-1}$). The goal was to evaluate how well the model could explain the time dependence of the observed growth and not to retrieve mass-based aerosol yields. As is shown in Fig. 18, the calculated production rate captures the variation in GR over the same example 3.5-day period highlighted in Fig. 10, 11, 12, and 15. During this period and for the remainder of the study the quality of the fit to the nighttime measurements





of GR was generally significantly better than that for the daytime measurements. Among the contributors to uncertainty in the daytime estimates are the poor constraint on OH· concentration and the exclusion of some VOC precursors such as the xylenes that react efficiently only with OH· and for which measured concentrations were very noisy and rarely above the detection limit of

the PTR-MS. Furthermore, no attempt was made to estimate the concentrations or impacts of unmeasured intermediate-volatile and semi-volatile organic compounds (IVOC and SVOC). The profile of the nighttime production rate was relatively insensitive to how it was partitioned between SOA production by $O_3$ and by $NO_3$·.

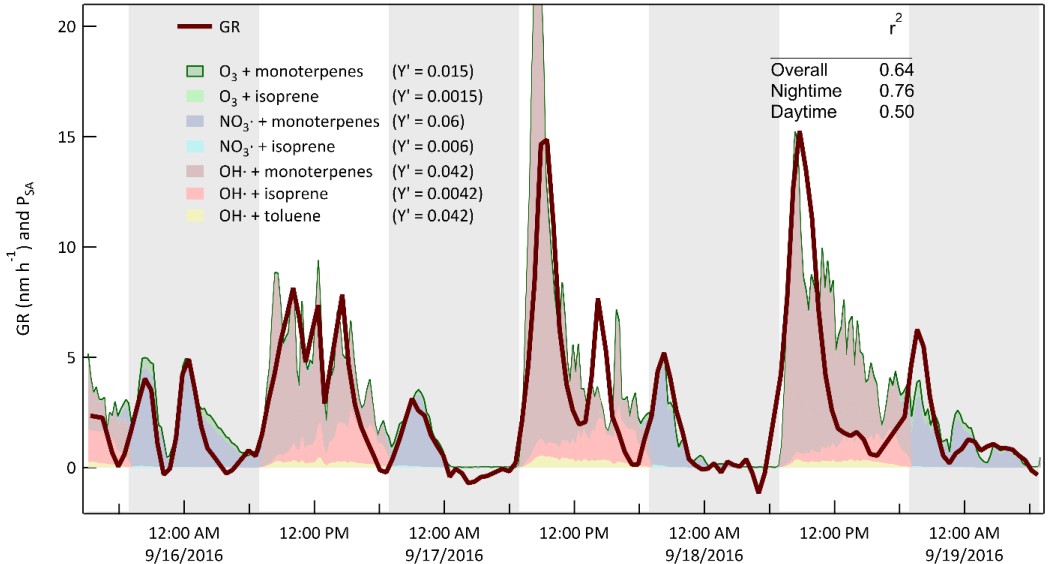

**Figure 18. Measured diameter growth rate (GR) and calculated secondary aerosol production rate ($P_{SA}$) for the same 3.5-day example period described above. Effective aerosol yields for the reactions considered were adjusted to minimize differences between the values of GR (in nm h$^{-1}$) and $P_{SA}$ (in μg m$^{-3}$ h$^{-1}$). The shaded bands represent nighttime.**

Though the characteristic bimodal growth rate pattern persisted throughout the two-month study, there was a significant shift in

the relative amplitudes of the daytime and nighttime maxima. The pattern change is evident in the contrast between the summer (8/15 – 9/21) and fall (9/22 – 10/14) hourly average GR profiles shown in Fig. 19. Unfortunately, explanation of the shift is difficult because measurement of the VOCs and all trace gases except $O_3$ ended on 9/22. At least some of the shift in nighttime particle growth is explained by a corresponding trend in nighttime $O_3$, with higher values more frequent later in the study. The average GR from 7:00 p.m. to midnight local time is correlated with the average $O_3$ mixing ratio for the same time interval, with an $r^2$ of 0.60

(Fig. S8). The correlation likely encompasses more rapid VOC oxidation due to the increased $O_3$ and to the increased $NO_3$· that forms from reaction of $O_3$ and $NO_2$. Additional insight into the short-term and seasonal variation in particle growth could come from longer term studies with more comprehensive gas-phase measurements that include monoterpene speciation.





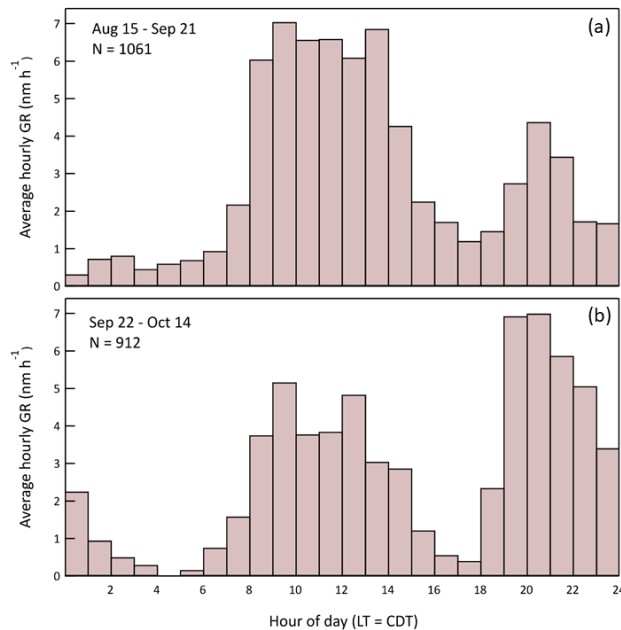

**Figure 19. Hourly average particle growth rates during the late summer (a) and early fall (b) portions of the study. A shift in the relative**
**importance of daytime and nighttime growth occurred between the start and end of the project.**

## 7 Conclusions

The Captive Aerosol Growth and Evolution (CAGE) chambers were characterized during a field study at a forested site outside of
Houston, TX. The CAGE chamber differs from most other Teflon® chambers in its portability, use of solar illumination to drive
photochemistry, rotation along its horizontal axis, and exchange of air with the surroundings through a permeable ePTFE
membrane. Chamber-ambient comparison for a range of measured organic and inorganic gases showed that concentrations in the
chambers were similar to those in the air surrounding them, with an effective exchange flow rate into and out of each chamber
through the permeable membrane of about 33 L min$^{-1}$. Results from those chamber-ambient characterization experiments were
used to validate a CSTR-0D model that uses measured concentrations of gases in ambient air to calculate concentrations of
measured and unmeasured species inside the chamber. Those calculated gas concentrations were subsequently used to estimate
production rates of low volatility compounds that contribute to secondary aerosol formation and particle growth. Narrow modes
of sub-0.1 μm diameter ammonium sulfate seed particles were repeatedly injected into the reaction chambers and their growth
rates measured 24 h day$^{-1}$ while they were exposed to conditions mirroring those outside. A mode of larger particles was maintained
in each chamber to provide stable surface area concentrations and, consequently, stable competition for condensable species.

Particle growth rate was measured continuously throughout the 2-month study. The observations that particle growth rate was
independent of particle size during periods when more than one mode was tracked simultaneously, and that the time of day growth
rates averaged over the study were all positive suggests that particle growth was caused mostly by low volatility species that
condensed irreversibly. The bimodality of the diel particle growth rate pattern results from late morning maxima from OH·
chemistry and evening maxima from O$_3$ and NO$_3$· chemistry. The diel pattern had a seasonal dependence that should be further
investigated. The temporal pattern of secondary aerosol production rate estimated using the CSTR-0D model was similar to that
of the measured particle growth rate, with an r$^2$ between the two time series of 0.64. Ongoing and future studies with CAGE



chambers are designed to quantify the sensitivities of particle growth to perturbations of ambient air caused by addition of one or more gases.

**Author contribution**

CLS and DRC prepared the manuscript with contributions from all co-authors. DRC, JMM, and CFM operated the CAGE chambers during the field study and analyzed the resulting dataset. MHE, JHF, RJS, SU, HWW, AATB, RJG, MT, SMK, and JLS participated in the field study and measured gas-phase and/or aerosol-phase concentrations and properties that were used in the characterization of the CAGE chambers.

**Competing interests**

The authors declare that they have no conflict of interest.

**Acknowledgments**

Funding support for operation of the CAGE chambers and the bioaerosol experiments was provided by the Defense Threat Reduction Agency through grant HDTRA1310184. Support for the University of Houston, Rice University, and Baylor University researchers was provided by the National Science Foundation through grants AGS-1552086 and AGS-1552077. We thank the Texas A&M Forest Service and the staff at the WG Jones State Forest headquarters for their assistance and for providing space for the chambers and trailers. We also thank the many undergraduate and graduate students at Texas A&M University that assisted in the construction and testing of the CAGE chambers described here and of their predecessors.

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
