# Peer review of "Captive Aerosol Growth and Evolution (CAGE) chamber system to investigate particle growth due to secondary aerosol formation"

_Atmospheric Measurement Techniques, 2020_

## Referee Comment (RC1) · Anonymous Referee #1 · 25 Dec 2020

GENERAL COMMENTS

In this manuscript the authors describe the design and evaluation of a portable chamber they have constructed primarily for studying aerosol growth under controlled conditions in outdoor air. It is 1 m3 and consists of a metal frame and Teflon film chamber that is permeable to gases but not particles so that particle growth can be studied under ambient conditions. The chamber walls are transparent to solar radiation, so that the ambient air photochemistry can be reproduced, and the chamber is slowly rotated to reduce losses of particles to the walls. The performance of the chamber was evaluated in a Texas field study by comparing measurements of ambient and chamber

concentrations of various trace gases and VOCs with predictions of a box model, and in general the agreement is excellent. Particle growth rates were measured over a few months by periodically adding seed particles to the chamber and measuring changes size distributions with an SMPS. The results provide valuable new measurements of the magnitude of growth rates and their dependence on particle size, which can provide insight into the growth mechanism, and diurnal and seasonal variations.

Overall, this is a very impressive new apparatus for studying gas and aerosol chemistry and particle growth under authentic atmospheric conditions. It is a major advance in the field and has applications beyond those described here. The manuscript is very clearly written and includes all the details and evaluation measurements one can hope for. I think it should be published in AMT after the following minor comments have been addressed.

SPECIFIC COMMENTS

1. Lines 282-287: The agreement of the curves in Figures 7-9 is obviously very impressive, but do the authors have any idea why at a few times the ambient concentrations significantly exceed chamber concentrations?

2. Lines 316-318: Are the ammonium sulfate seed particles in the chamber dry or deliquesced? Deliquesced ammonium sulfate particles will generally have very low pH ($\sim$1 or so) due to evaporation of ammonia. If the particles are deliquesced do the authors have any idea what ambient ammonia concentrations were? The pH could be estimated using E-AIM, for example. The nature of the seed could have a significant impact on SOA formation and growth via aqueous phase chemistry and acid catalysis. The authors might discuss this issue and offer suggestions on the best seeds to use, depending on measurement goals.

3. Line 388: Can the authors describe what steps were taken to minimize wall charging?

4. Are particle wall losses due mostly to diffusion?

5. Line 417+: I suggest the authors provide some discussion of how gas-wall partitioning of VOC reaction products to the chamber walls can influence measurements of aerosol growth rates and products. It is now well established (e.g., Matsunaga and Ziemann, AST, 2010; Krechmer et al., EST, 2016) that this process has a significant impact on SOA formation and that equilibrium is reached in Teflon chambers on timescales of ∼10 min and probably less in this small chamber.

TECHNICAL COMMENTS

1. Line 23: Should be "membrane is".

---

## Referee Comment (RC2) · Anonymous Referee #3 · 27 Dec 2020

The authors describe a new type of enclosure chamber which allow to observe particle growth by seeding a real mix of ambient atmospheric air exposed to outside light conditions. Long residence time of the seed particles is provided by rotating the cylindric chambers around the center axis which reduces gravitational settling and convective mixing. Gas exchange is provided by permeation through a membrane, such that the injected seed particles are not subject to flush out. The latter concept was also applied before (QUALITY camber). The presented chambers provide still a quite substantial progress compared to the previous versions. The results are overall presented well and look promising. And I think the manuscript shows indeed proof of concept. However, from the descriptions in the manuscript, I could not understand all features of

the new chamber. Some important information is missing. The manuscript is within the range of AMT and could be published after the authors addressed the comments below.

Comments

In the manuscript the authors seem not to distinguish between mixing time and exchange time (e.g. line 22, abstract).

In my opinion there is a difference. If the mixing time is short compared to the residence time, the chamber will be always well mixed, even when the boundary conditions are (slowly) changing. Isn't fast mixing a prerequisite for treatment of the chamber as a continuously stirred tank reactor? If the chamber is not mixed fast, in how far is the sampling from one (or two?) points inside the chamber representative for the processes in and state of the chamber? Please describe, how the mixing within the volume of the chamber is assured, the mixing time and mixing behavior in the CAGE chamber?

A bit related to the previous point. You are taking samples for measurements from the chamber by two DMAs, APS, and AMS. How large is the total sample flow? I guess it is significant compared to the residence time? How much does this sample flow affect the exchange time? Does the sample flow enhance the mixing in the chamber?

Was the sheath for the DMA(s) taken from the chamber? If not in how far could not using sheath air from the chamber and drying of the aerosol have affected the observation that only condensation took place (that growth rate always > 0)?

Line 126: I have difficulties to understand how the sampling is done. Is the centers axis a tube which serves as a sampling line at the same time? Is that little perpendicular extension in Figure S2 to the very right one of the sampling tubes? Do you sample from both tubes or are you using one for sampling and the other for injection of seed particles? Do all instruments sample from the same point?

Line 323: I think you use "Dp" as size parameter for the size distributions and as mode

diameter. I suggest using two different notations and call latter Dmode, or so.

Line 423 and Figure 15. I can see that the traces look different, but also different quantities are plotted. Please, extent the description why exactly we can see from this Figure the difficulty in determining secondary aerosol production from mass measurements. I guess you have all information to calculate the mass production from your growth rates using an average particle density (from AMS measurements)? Or I don't understand at all what you want to address.

Line 459: Since O3 is substantially lower in the night than during day time, ozonolysis ([VOC]*[O3]) is probably not driving the night time maximum.
* * *

---

## Referee Comment (RC3) · Anonymous Referee #2 · 30 Dec 2020

The Captive Aerosol Growth and Evolution (CAGE) chamber system is a novel idea in order to investigate particle growth related to secondary aerosol formation. The use of a gas-permeable ePTFE membrane ensures that the gas composition of the chambers matches the ambient one. The manuscript is well written and provides a detailed analysis of the design and the evaluation of their performance.

Specific comments: One key aspect not shown is related to temperature measurements of the air inside the chambers. It is clearly very difficult to maintain the same temperature between the two chambers when one covered to prevent sunlight and the other uncovered during the field deployments. The blower used to circulate air through

the exterior of the chambers minimizes such effects to some extent. Where there any temperature measurements conducted either from the external of the chambers or from inside of the chambers or from the sampled air (event the readings from the instrumentation) and if yes how different were the two chambers (the covered vs the uncovered). The temperature difference might affect the wall loss profiles for the two chambers and the reaction rates.

Page 4, line 127: radial O-rings. What material are they made from.

Page 5 line 145: Can you provide the related data in the SI from the spectroradiometer for the reduction.

---

## Author Comment (AC1) · 16 Mar 2021

GENERAL COMMENTS: In this manuscript the authors describe the design and evaluation of a portable chamber they have constructed primarily for studying aerosol growth under controlled conditions in outdoor air. It is 1 m3 and consists of a metal frame and Teflon film chamber that is permeable to gases but not particles so that particle growth can be studied under ambient conditions. The chamber walls are transparent to solar radiation, so that the ambient air photochemistry can be reproduced, and the chamber is slowly rotated to reduce losses of particles to the walls. The performance of the chamber was evaluated in a Texas field study by comparing measurements of ambient and chamber concentrations of various trace gases and VOCs with predictions of a box model, and in general the agreement is excellent. Particle growth rates were measured over a few months by periodically adding seed particles to the chamber and measuring changes size distributions with an SMPS. The results provide valuable new measurements of the magnitude of growth rates and their dependence on particle size, which can provide insight into the growth mechanism, and diurnal and seasonal variations. Overall, this is a very impressive new apparatus for studying gas and aerosol chemistry and particle growth under authentic atmospheric conditions. It is a major advance in the field and has applications beyond those described here. The manuscript is very clearly written and includes all the details and evaluation measurements one can hope for. I think it should be published in AMT after the following minor comments have been addressed.

SPECIFIC COMMENTS:

1. Lines 282-287: The agreement of the curves in Figures 7-9 is obviously very impressive, but do the authors have any idea why at a few times the ambient concentrations significantly exceed chamber concentrations?

We believe the question is regarding the spikes in ambient concentration rather than the periods when there is a difference that is explained by chemical loss (e.g., of isoprene) in the chamber. As is most clearly seen in the center satellite image in Figure 4, the site was in a forested area, but still close to a highway and only a few km from a major interstate. Emissions from those and other local sources are believed to be responsible for the spikes in NO and at least some other species. Those spikes are smoothed out in the chamber because of the ~30 min exchange time with the surrounding ambient air.

2. Lines 316-318: Are the ammonium sulfate seed particles in the chamber dry or deliquesced? Deliquesced ammonium sulfate particles will generally have very low pH (~1 or so) due to evaporation of ammonia. If the particles are deliquesced do the authors have any idea what ambient ammonia concentrations were? The pH could be estimated using E-AIM, for example. The nature of the seed could have a significant impact on SOA formation and growth via aqueous phase chemistry and acid catalysis. The authors might discuss this issue and offer suggestions on the best seeds to use, depending on measurement goals.

The seed particles were dried prior to injection. However, the ambient and chamber RH often exceeded the deliquescence RH of ammonium sulfate at night. Thus, the particles were sometimes dry and sometimes aqueous, and sometimes there was undoubtedly a mixture of dry and aqueous particles in the chamber at the same time. We appreciate the importance of aerosol composition, phase state, and

pH on particle phase chemistry and SOA formation.  Some of our near-future experiments with the chambers are intended to examine the influence of seed particle composition and water content.  For the experiments described in this manuscript we wanted to focus on variability with time-of-day and between days and, therefore, we tried to keep everything else unchanged, including the composition of the seed particles.  Though it is not conclusive, some evidence that seed particle composition did not have a significant impact comes from the similarity in the growth rates of particles that formed from nucleation and those that were initially pure ammonium sulfate, as shown, for example, in Figure 12.

We added the following sentence to Section 5 immediately after stating that the seed particles were composed of ammonium sulfate.

"Future studies are planned to evaluate the sensitivity of particle growth to the composition of the seed particles."

3. Line 388: Can the authors describe what steps were taken to minimize wall charging?

Following assembly of the chamber but prior to putting the acrylic sides on, concentrated bipolar ions were generated with a Po-210 source and directed towards the outside of the FEP as the chamber rotated.  The acrylic sides were then installed, which prevented any further contact that could result in static charge.  Some charge was likely removed at night when the RH inside and outside of the chamber was high.  Towards the end of the campaign we also experimented with sliding a Po-210 source back and forth on a track underneath the chamber as it rotated.  Since then we have stopped using Po-210 sources and rely instead on bipolar corona discharge ionizers, which are not quite as effective (or balanced), but obviously much easier to ship and use.

4. Are particle wall losses due mostly to diffusion?

We believe that diffusion is a large contributor to the loss of all but the supermicron particles.  Unfortunately, it is difficult to determine the relative importance of diffusion and of electrostatic loss based on our results.  The size range spanned by the tracked mode particles was not large enough to allow us to infer the dominant mechanism(s) based on any size-dependence of the loss.  Furthermore, any attempt to quantify the size dependence would be complicated by time-of-day dependent variability in convective mixing (as is evident in Figure 15a) and day-of-experiment variability in things like static charge.  Nevertheless, the observation that particle lifetime in these chambers is comparable to that in much larger ones, as summarized in Table 1, suggests that electrostatic loss is comparatively less important than in other Teflon chambers (making diffusion comparatively more important).

5. Line 417+: I suggest the authors provide some discussion of how gas-wall partitioning of VOC reaction products to the chamber walls can influence measurements of aerosol growth rates and products. It is now well established (e.g., Matsunaga and Ziemann, AST, 2010; Krechmer et al., EST, 2016) that this process has a significant impact on SOA formation and that equilibrium is reached in Teflon chambers on timescales of ~10 min and probably less in this small chamber.

We added the following brief discussion towards the end of Section 4:

"No attempt was made to account for gas-wall partitioning of VOC reaction products, despite recognition that such partitioning is significant and can complicate interpretation of results from Teflon chambers (Matsunaga and Ziemann, 2010; Krechmer et al., 2016). For species that partition reversibly to the walls, the impact may be only an increase in the ~30 min effective chamber-ambient exchange time by an amount comparable to the ~10 min time scale for reaching equilibrium for photochemically generated oxidation products as reported by Krechmer et al. (2016)."

TECHNICAL COMMENTS 1. Line 23: Should be "membrane is".

We fixed this.  Thank you.

---

## Author Comment (AC2) · 16 Mar 2021

GENERAL COMMENTS: The authors describe a new type of enclosure chamber which allow to observe particle growth by seeding a real mix of ambient atmospheric air exposed to outside light conditions. Long residence time of the seed particles is provided by rotating the cylindric chambers around the center axis which reduces gravitational settling and convective mixing. Gas exchange is provided by permeation through a membrane, such that the injected seed particles are not subject to flush out. The latter concept was also applied before (QUALITY camber). The presented chambers provide still a quite substantial progress compared to the previous versions. The results are overall presented well and look promising. And I think the manuscript shows indeed proof of concept. However, from the descriptions in the manuscript, I could not understand all features the new chamber. Some important information is missing. The manuscript is within the range of AMT and could be published after the authors addressed the comments below.

SPECIFIC COMMENTS: In the manuscript the authors seem not to distinguish between mixing time and exchange time (e.g. line 22, abstract).

In my opinion there is a difference. If the mixing time is short compared to the residence time, the chamber will be always well mixed, even when the boundary conditions are (slowly) changing. Isn't fast mixing a prerequisite for treatment of the chamber as a continuously stirred tank reactor? If the chamber is not mixed fast, in how far is the sampling from one (or two?) points inside the chamber representative for the processes in and state of the chamber? Please describe, how the mixing within the volume of the chamber is assured, the mixing time and mixing behavior in the CAGE chamber?

This is an excellent question and one for which we do not have an excellent answer.  We feel there is some optimal amount of mixing within the chamber – too much and gas and, especially, particle wall losses will be excessive, and too little and there will be significant gradients in gas phase concentrations, especially for more reactive species.  It has been our opinion that mixing in the chamber is typically on the high side of that optimum, meaning that we have sought to minimize it.  Of course, the balance between the two effects depends on the nature and objectives of the experiment.  We did not perform any experiments that would allow us to separate the mixing time from the exchange time.  For now, we have replaced "mixing" with "exchange" for both instances in the text where it was mistakenly used (abstract and original line 266).

A bit related to the previous point. You are taking samples for measurements from the chamber by two DMAs, APS, and AMS. How large is the total sample flow? I guess it is significant compared to the residence time? How much does this sample flow affect the exchange time? Does the sample flow enhance the mixing in the chamber?

The APS was configured with separate sample and sheath flows, such that it pulled only 1 L/min from the chamber (and not the total 5 L/min including the sheath flow rate).  The SMPS had a sample flow rate of about 2.1 L/min.  Adding in the much smaller AMS flow rate results in an overall sample of about 3.5 L/min.  However, a sampling sequence was used to minimize extraction from either chamber.  Specifically, the repeated sequence used during the study described in the manuscript was Chamber A

→ Chamber B → 4 x ambient.  The result is that sample was extracted from each chamber only 1/6$^{th}$ or 16.7% of the time, meaning that the effective withdrawal rate was only about (3.5 L/min) x 0.167 = 0.58 L/min.  The particle injection flow rate was about 3 L/min.  The particle injection frequency and duration varied, with an average of around 20 to 30 min per day, resulting in an average chamber air displacement rate of about 0.05 L/min.  The corresponding loss time constant is 1000 L / 0.63 L/min = 1,600 min or about 26 h.  Of the particle size categories considered, the maximum residence time was about 8.2 h.  Thus, extraction contributes to the loss, but is less important than other contributors such as diffusion and electrostatic loss.  The reported particle residence times of the three size categories of 6.0 h (small particles), 8.2 h (medium particles), and 3.9 h (large particles) would be approximately 7.8 h, 11.9 h, and 4.6 h, respectively, without any sample or injection flows.

The intermittent sampling approach was clarified in the manuscript by revising the following sentence in Section 2.5 of the original manuscript:

"The instrumentation was configured to sample from both the inside of each of the two chambers and ambient air."

To now be:

"The instrumentation was configured to sample from both the inside of each of the two chambers and ambient air, with a repeated sampling sequence of Chamber A → Chamber B → 4 x ambient, such that sample was extracted from each chamber only 1/6th of the time in order to minimize the loss rate of the captive particles."

Was the sheath for the DMA(s) taken from the chamber? If not in how far could not using sheath air from the chamber and drying of the aerosol have affected the observation that only condensation took place (that growth rate always > 0)?

The sheath flow in the SMPS DMA was configured in a recirculation loop.  Thus, all of the air in the DMA came from the same source.  The interior volume of the DMA is about 1.6 L.  The filter and blower enclosure in the recirculation loop probably bring the total volume up to about 2 L.  As noted above, the SMPS sample flow rate was about 2.1 L/min, resulting in a turnover time constant for the air in the DMA of about 1 min, which is short compared with the 5 min sampling time used.  If carryover from a previous measurement (e.g., from Chamber A when sampling from Chamber B) had a significant impact, it would be reflected in differences between the up and down scans in the SMPS measurements, which was not observed.  It also seems unlikely that the gas phase would result in a positive bias in measured growth rate because low volatility species would likely be depleted in the stainless sampling line or in the DMA itself.

Line 126: I have difficulties to understand how the sampling is done. Is the centers axis a tube which serves as a sampling line at the same time? Is that little perpendicular extension in Figure S2 to the very right one of the sampling tubes? Do you sample from both tubes or are you using one for sampling and the other for injection of seed particles? Do all instruments sample from the same point?

Particles are injected into one side of the chamber and are extracted from the other side. The two tubes that are perpendicular to the axle in Figure S2 are the sampling and injection ports. (The third, which is at about a 45 degree angle was intended for gas sampling, but was not used.) Those 0.95 cm OD tubes are bent 90 degrees inside the hollow axle and then continue through opposite ends of the axle where they are sealed into fixed unions using radial o-rings. Thus, the injected or sampled flow remains within a single 0.95 cm OD stainless tube from outside of the chamber to the end of one of those perpendicular tubes.

Line 323: I think you use "Dp" as size parameter for the size distributions and as mode diameter. I suggest using two different notations and call latter Dmode, or so.

We have changed instances of $D_p$ used to describe the fitted mode diameter with $D_{mode}$, including those in Figures 11 and 13.

Line 423 and Figure 15. I can see that the traces look different, but also different quantities are plotted. Please, extent the description why exactly we can see from this Figure the difficulty in determining secondary aerosol production from mass measurements. I guess you have all information to calculate the mass production from your growth rates using an average particle density (from AMS measurements)? Or I don't understand at all what you want to address.

Our objective in including that figure was simply to argue that it can be challenging to quantify the SA production rate from measurements of mass (or volume) concentration as would typically be necessary if a polydisperse seed aerosol were instead injected. For a typical batch chamber, the rate of SA mass production can be controlled to be greater (or much greater) than the rate at which SA already on particles is lost to the walls. Within the CAGE chambers, the SA production rate is comparatively low and highly variable. The result is that the noise caused by variability in the wall loss rate is often greater than the signal resulting from added SA to the particles in the chamber. This is reflected in the noisy curve showing the production rate inferred from the AMS measurements.

Line 459: Since O3 is substantially lower in the night than during day time, ozonolysis ([VOC]*[O3]) is probably not driving the night time maximum.

As shown in Figure 17, the average rate of decay of the $O_3$ concentration at night is comparable to that calculated for $NO_3\cdot$ (which is, of course, not surprising because $O_3$ reaction produces $NO_3\cdot$). With our model, the sort of production rate estimate shown in Figure 18 was not very sensitive to whether the $O_3$ reacted with the precursors or the $O_3$ produced $NO_3\cdot$, which reacted with the precursors. Though $O_3$ decreases rapidly in the evening, the concentrations of precursor VOCs (especially monoterpenes) increase rapidly, with the result that the product of the concentrations often reaches a maximum around the time of the peak in particle growth rate.

---

## Author Comment (AC3) · 16 Mar 2021

GENERAL COMMENTS: The Captive Aerosol Growth and Evolution (CAGE) chamber system is a novel idea in order to investigate particle growth related to secondary aerosol formation. The use of a gas-permeable ePTFE membrane ensures that the gas composition of the chambers matches the ambient one. The manuscript is well written and provides a detailed analysis of the design and the evaluation of their performance.

SPECIFIC COMMENTS: One key aspect not shown is related to temperature measurements of the air inside the chambers. It is clearly very difficult to maintain the same temperature between the two chambers when one covered to prevent sunlight and the other uncovered during the field deployments. The blower used to circulate air through the exterior of the chambers minimizes such effects to some extent. Where there any temperature measurements conducted either from the external of the chambers or from inside of the chambers or from the sampled air (event the readings from the instrumentation) and if yes how different were the two chambers (the covered vs the uncovered). The temperature difference might affect the wall loss profiles for the two chambers and the reaction rates.

During the experiments described in the manuscript, we did not record temperature inside the chamber enclosures.  (Since then we have added multiple sensors and have installed small, variable speed air conditioners to control the temperature).  We did, however, install simple digital-display temperature sensors inside the chamber enclosures and on the outside, just below the chamber enclosures (where it was always shaded).  We did our best to put ventilated light covers around the sensors inside the enclosures and wrapped the probes with Teflon tape to minimize bias from solar heating.  We found that the temperature inside the uncovered enclosure was about 3 °C higher than that outside in the middle of the day.  Of course, at night there was little difference.  The temperature difference was much lower with the covered chamber.  However, the only measurements from the "covered" chamber that were used in the manuscript were those from the period when the chamber-ambient characterization experiment was conducted, during which the "covered" chamber was not covered (lines 148 and 208).

Line 148: "With the exception of the results from the chamber-ambient characterization experiment described below, only measurements from Chamber A will be described here."

Line 208: "Unlike the rest of the 2-month study, Chamber B was uncovered for these experiments in order to assess the chamber-to-chamber consistency."

We added a brief mention of the steps taken to minimize heating in Section 2.4:

"The ventilation air flow created by the blower, together with the use of light reflective materials and coatings, helps minimize heating of the chamber above the surrounding temperature during daytime."

Page 4, line 127: radial O-rings. What material are they made from.

We at first used FEP-encapsulated viton o-rings.  Unfortunately, those did not last long.  We then switched to PTFE o-rings, which probably didn't seal as well, but were much more resilient.

Page 5 line 145: Can you provide the related data in the SI from the spectroradiometer for the reduction.

The spectroradiometer was not used for this comparison. Instead, a digital-display, total UV sensor was positioned inside and then outside of the enclosure of the covered chamber. It was noted that the covers reduced the intensity by 99%, but the values were not recorded. It is likely that the relative reduction varied over the course of the campaign as the reflective covers got dirty (possibly increasing the efficiency) and less rigid (likely decreasing the efficiency). Regardless, as noted above, no data from the covered chamber was used in this manuscript.